# Terahertz field effect in a two-dimensional semiconductor

Tomoki Hiraoka [1] ✉, Sandra Nestler[2], Wentao Zhang [1], Simon Rossel[1], Hassan A. Hafez [1], Savio Fabretti [1], Heike Schlörb[2], Andy Thomas[2,3] & Dmitry Turchinovich [1] ✉

Layered two-dimensional (2D) materials offer many promising avenues for advancing modern electronics, thanks to their tunable optical, electronic, and magnetic properties. Applying a strong electric field perpendicular to the layers, typically at the MV/cm level, is a highly effective way to control these properties. However, conventional methods to induce such fields employ electric circuit - based gating techniques, which are restricted to microwave response rates and face challenges in achieving device-compatible ultrafast, sub-picosecond control. Here, we demonstrate an ultrafast field effect in atomically thin $MoS_2$ embedded within a hybrid 3D-2D terahertz nanoantenna. This nanoantenna transforms an incoming terahertz electric field into a vertical ultrafast gating field in $MoS_2$, simultaneously enhancing it to the MV/cm level. The terahertz field effect is observed as a coherent terahertz-induced Stark shift of exciton resonances in $MoS_2$. Our results offer a promising strategy to tune and operate ultrafast optoelectronic devices based on 2D materials.

Two-dimensional (2D) layered materials, with their atomic-scale thickness and remarkable electronic, optical, and mechanical properties, offer significant potential across a broad range of scientific and technological fields, from classical and quantum computing to photodetection, energy harvesting, spintronics, memristor technology, and the discovery of new quantum phases[1–8]. The primary advantage of layered 2D materials is the broad tunability of their key physical properties, which stem from microscopic interactions within and between the material layers[9,10], as well as between the material and its environment[11–13].

One very effective way to control these intra- and interlayer interactions, and hence the physical properties of 2D materials, is via field effect caused by a strong electric field perpendicular to the material layers[14]. Field effect enables precise control of conductivity in field-effect transistor channels, as well as of the efficiency and detection range of 2D photodetectors, the generation of ultrahigh-frequency signals, and other processes critical to modern technology[15–18]. Moreover, it is crucial for exploration of the fundamental physics of 2D materials, allowing control over bandgaps[14,19–21] and excitonic states[22–28], insulator-metal transition[16,20], structural phase transitions[29], quantum phase transition in moiré systems[8] and more. Due to their atomic thickness, achieving significant control in 2D materials requires very strong electric fields, typically of the order of MV/cm[16,17,20–28].

Technologically, the field effect is realized by electrostatic gating. So far, gating of 2D materials has been limited to static DC-based[16–29] or microwave-based[30–32] methods, thus lacking the technologically-relevant ultrafast, sub-picosecond control capabilities. The key challenge here lies in the practical way of applying the terahertz (THz) gating electric signal of MV/cm strength to the 2D material in the direction perpendicular to its layers. Here, we demonstrate a pronounced THz field effect in a model 2D semiconductor $MoS_2$[9,10,16,21–23,25,28]. For this, we introduce a hybrid 3D-2D nanoantenna structure that receives the broadband THz electromagnetic signal

[1]Fakultät für Physik, Universität Bielefeld, Bielefeld, Germany. [2]Leibniz-Institut für Festkörper- und Werkstoffforschung, Helmholtzstraße 20, Dresden, Germany. [3]Institut für Festkörper- und Materialphysik, Technische Universität Dresden, Haeckelstraße 3, Dresden, Germany. ✉e-mail: tomoki.hiraoka@riken.jp; dmtu@physik.uni-bielefeld.de

from the free space and converts it into a vertical gating field in the MoS$_2$, simultaneously enhancing it to a MV/cm level. Using time-resolved optical measurements, we observe the field effect as a coherent THz-induced Stark shift of characteristic exciton resonances in MoS$_2$. Therefore, our approach enables efficient ultrafast 2D (opto) electronic technology controlled by the THz fields, as well as experiments in fundamental investigations of 2D materials, where THz field effect is essential.

## Results

### A hybrid 3D-2D nanoantenna for THz gating

The design of our hybrid 3D-2D nanoantenna for THz gating of 2D materials is presented in Fig. 1. The antenna consists of two gold electrodes, top and bottom, vertically separated by an Al$_2$O$_3$ dielectric spacer layer. As shown in Fig. 1a, b, the electrodes are horizontally displaced such that they only overlap in the middle section of the antenna, which has lateral dimensions of 10 μm x 10 μm. In the top view projection, our antenna has a bowtie dipole shape, which enables efficient incoupling of broadband free-space THz fields to its electrodes and strong local field enhancement in the sub-wavelength antenna gap[33,34]. The entire antenna structure is deposited on a glass substrate. The 150-nm-thick Al$_2$O$_3$ dielectric spacer layer was grown by atomic-layer deposition[35,36] between the electrode planes in order to prevent the dielectric breakdown in the antenna in strong THz fields. Figure 1c shows the microscope optical image of our hybrid 3D-2D nanoantenna. Several such devices were fabricated in two batches (below referred to as α and β - batches) and tested in this study. For the details see "Methods" Section 1,2, and the Performance Variation Among Fabricated Devices section below.

The application of a THz electromagnetic field $F_{x,\,in}$, vertically illuminating the entire antenna structure and polarized along its electrodes, leads to a THz-field-driven current in the top and bottom electrodes. Due to their vertical separation and only partial horizontal overlap, this creates an electric polarization in the middle section of

the antenna in the vertical direction. If a 2D material is placed in this region with layers parallel to the electrodes, it will experience the vertical gating THz field $F_z$, as shown in Fig. 1d. Besides conversion of the horizontally polarized incident THz field into the vertical gating field, our antenna also provides a significant field enhancement in the vertical direction. Finite difference time domain (FDTD) simulations of our antenna allow for the calculation of the gating THz field $F_z$ from the experimentally measured incident THz field $F_{x,\,in}$ at the antenna gap position. This resulted in a calculated field enhancement factor of the antenna of $|F_z/F_{x,\,in}| = 13.3$, as shown in Fig. 1e. This field enhancement factor resulted in a maximum gating THz field $|F_z|$ in excess of 1 MV/cm applied to a MoS$_2$ sample, an estimate confirmed in our measurements. We note, however, that our experiments do not allow for the direct measurement of $|F_z/F_{x,\,in}|$. Therefore, we present all the relevant THz-field dependent parameters in this work as a function of incident THz field $F_{x,\,in}$, a directly measured quantity. The details of the calculation of the gating field $F_z$ from the measured incident THz field $F_{x,\,in}$ can be found in Supplementary Notes 1, while the experimental estimate of the gating field strength is presented in the Discussion section below.

Several single-crystalline MoS$_2$ flakes, prepared by mechanical exfoliation, each containing 3-4 layers, were studied in this work. In our experiments, we used optical probing to demonstrate the THz gating effect on the MoS$_2$ flakes, which are positioned horizontally in the middle field-enhancement section of the antenna, as shown in Fig. 1a, b, d, e. In order to ensure optical access to the MoS$_2$, the bottom electrode of our hybrid 3D-2D nanoantenna was made 8 nm thin and, therefore, optically semitransparent[37], whereas the top electrode thickness was 25 nm and 50 nm for the devices in α- and β- batches, respectively. As shown in Fig. 1d, the optical probe pulse enters the structure through the bottom electrode, travels through the spacer region interacting with the MoS$_2$ flake, gets back-reflected off the thick top electrode, and is then directed to the optical detection. We note that, in such an arrangement, the relatively low thickness of the bottom

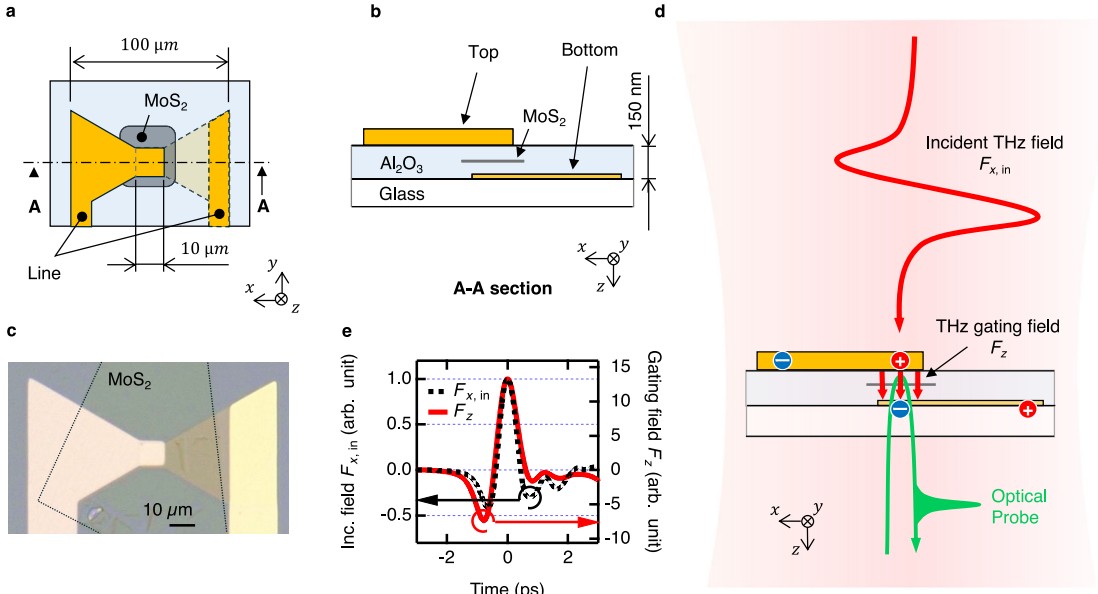

**Fig. 1 | Hybrid 3D-2D THz nanoantenna. a** Schematic top view of the device around the bowtie antenna part. **b** Schematic side view of the device (A-A section of **a**). **c** Microscope image of a fabricated device. The dotted line represents the MoS$_2$ flake. The scale bar corresponds to 10 μm. **d** Schematic of the THz pump-optical probe experiment. The red and green curved arrow represents incident THz field and optical probe, respectively. The straight arrows in the gap of the antenna represents the enhanced THz gating field. The blue and red circles represent negative and positive charge density change. **e** Simulated field enhancement. Black dashed line: measured incident THz field $F_{x,\,in}$. Red solid line: gating THz field in the antenna gap $F_z$, calculated numerically from the measured incident THz field $F_{x,\,in}$. The calculated field enhancement factor is $|F_z/F_{x,\,in}| = 13.3$. Both vertical axes in (**e**) are on the same arb. unit scale. The traces $F_{x,\,in}$ and $F_z$ are horizontally offset to show the peaks of both signals at 0 ps.

gold electrode of 8 nm was chosen to achieve both optical transparency as required for the optical probing and reasonably high conductance[38] as required for efficient electrical gating. In a situation where no optical access to the 2D material is required, using thicker gold for both electrodes will lead to a higher conductance and, hence, to an even larger field enhancement factor $|F_z/F_{x,\,in}|$. The design of our antenna enables efficient incoupling of incident free-space THz signal with the bandwidth of at least 0.1 – 2.5 THz and its conversion into the field-enhanced THz gating field without any loss of bandwidth (see Supplementary Notes 1).

## THz pump-optical probe measurements

As a probing mechanism for the THz gating effect, we used time-resolved optical spectroscopy of characteristic exciton resonances in MoS₂. For this, a THz pump-optical probe (TPOP) experiment[39] was set up, as schematically shown in Fig. 1d. The pump THz field was generated via tilted pulse front optical rectification of a 2 mJ, 800 nm laser pulse with 100 fs duration in a lithium niobate crystal[40,41]. This produced a broadband single-cycle pulse of the THz field with a frequency range of 0.2–2.5 THz and a center frequency of 0.4 THz, propagating into the free space. The THz beam was then focused and directed at our antenna at normal incidence, leading to the maximum THz field strength of $|F_{x,\,in}| = 104$ kV/cm, as measured at the position of the antenna structure using free-space electrooptic sampling[42]. In order to prevent the potential breakdown of the studied devices, we made sure that the field-enhanced gating THz signal $|F_z|$ does not exceed the estimated values of 1.2–1.5 MV/cm by properly attenuating the incident THz field. For the gating field estimate, see the Discussion section. The incident THz field was polarized along the bowtie antenna structure, as shown in Fig. 1a–d. The diameter of the THz focus spot (1/$e$ width for the field) is estimated as 1 mm at the center frequency of 0.4 THz, which is much larger than the antenna size. A broadband optical probe pulse was produced via supercontinuum broadening of another 800 nm, 100 fs laser pulse with 3 μJ energy in a sapphire slab of 4 mm thickness[43]. The generated optical supercontinuum signal was then passed through a bandpass filter, resulting in an optical probe pulse with 900 fJ energy and covering the photon energy range of ca. 1.7–2.3 eV as required for our measurements. The probe pulse chirp correction is addressed in Methods Section 4 and Supplementary Notes 3.

The THz pump field and the optical probe pulse were directed at the antenna from the side of the top and bottom electrodes, respectively, as shown schematically in Fig. 1d. The optical probing was performed in reflection in a confocal configuration using the ×10 magnification microscope objective, where the optical probe was focused on the field-enhancement region of the antenna, subject to the gating field $F_z$. For the analysis of the optical probe reflectance of our structure, an imaging CCD spectrometer was used to simultaneously record the spectra of the incident and reflected optical probe pulses. As a result, we were able to measure the calibrated broadband optical reflectance spectra of the field-enhancement region of our antenna as a function of the time delay between the incident THz pump field and the broadband optical probe pulse. All our experiments were performed at room temperature. See the Methods Section 3 for the details of our experimental setup.

At first, we characterized the response of a reference structure, a hybrid 3D-2D nanoantenna *without* the MoS₂ sample, to the incident THz field. The results are shown in Fig. 2a, c, f. A measured static optical reflectance spectrum $R_{st}$ of the reference structure without applied THz field is shown in Fig. 2a. This spectrum has a simple convex shape originating from the reflectance of gold electrodes and multi-reflection in the antenna gap, as confirmed by our optical simulations (see Supplementary Notes 2-1). Figure 2c shows the temporal evolution of the differential optical reflectance spectrum $\Delta R/R$ of the reference antenna subject to the incident THz field $F_{x,\,in}$ with a peak

amplitude of 104 kV/cm, shown in Fig. 2e. Given the field enhancement factor of $|F_z/F_{x,\,in}| = 13.3$, this corresponds to an estimated peak gating field in the gap region of the antenna of $F_z = 1.4$ MV/cm. The THz field leads to a small differential reflectance modulation of the reference antenna structure of $|\Delta R/R| < 1\%$, spectrally spread over the entire range of probe photon energy of 1.75–2.15 eV. This modulation is several picoseconds long and does not show any temporal structure present in the incident THz waveform, which is shown in Fig. 2e. We attribute this response to the long-living THz-induced modulation of the refractive index of the spacer layer and/or of the reflectance of the thin gold electrode in the field-enhancement region of the antenna. See Supplementary Notes 2-2 for details. In Fig. 2f, an absolute optical reflectance spectrum $R$ of a reference antenna structure is shown as a function of the THz pump-optical probe delay. It does not demonstrate any significant temporal modulation, and its overall shape resembles that of the static optical reflectance spectrum of the reference antenna without the THz field, shown in Fig. 1a.

After the reference antenna without the MoS₂ sample was characterized, we proceeded with the measurements on an actual device containing the MoS₂. Figure 2b, d, g shows the static optical reflectance spectrum $R_{st}$ without the THz field, the time-dependent differential reflectance spectrum $\Delta R/R$, and the time-dependent absolute reflectance spectrum $R$, respectively, of one of our manufactured devices - an antenna containing the 3 layer (3 L) flake of MoS₂ in its field-enhancement region. The static optical reflectance spectrum of the antenna with 3 L MoS₂ (Fig. 2b) now shows two pronounced dips around the probe photon energies of 1.89 eV and 2.03 eV. These spectral dips correspond to optical absorption due to well-known A- and B-exciton transitions in MoS₂[9]. When this device was subject to the incident THz field $F_{x,\,in}$ with peak amplitude of 104 kV/cm (see Fig. 2e, h), corresponding to a maximum estimated gating field of $|F_z| = 1.4$ MV/cm, a very clear spectral and temporal modulation was observed in the differential reflectance spectrum $\Delta R/R$ of the antenna loaded with MoS₂ (Fig. 2d). Spectrally, these THz-induced modulations are concentrated around the A- and B-exciton transitions. Temporally, these modulations resemble the structure of the incident THz field, shown in Fig. 2e. The time evolution of the absolute optical reflectance spectrum $R$ of the antenna with MoS₂, shown in Fig. 2g, shows modulation of spectral positions of both A- and B-exciton lines, time-correlated with the oscillation of the incident THz field $F_{x,\,in}$, shown in Fig. 2h. We have therefore observed a clear effect of THz gating of the MoS₂ sample as spectral and temporal modulation of its characteristic A- and B-exciton transitions, an effect also confirmed by our optical simulations (see Supplementary Notes 2). When the antenna was rotated by 90° with respect to the THz field polarization to avoid the incoupling of the incident THz field into the antenna structure, no measurable modulation of optical probe reflectivity was observed in our devices. In the following, we will perform a detailed analysis of these THz-induced optical modulations of MoS₂.

We note that the time evolution of the incident THz field $F_{x,\,in}$ and the time evolution of the optical reflectance spectra $R$ and $\Delta R/R$ were measured separately, and the precise relative timing offset between them could not be established experimentally. Therefore, in the data presentation, we chose the timing such that the maximal modulation of optical reflectance of MoS₂ would correspond to the strongest THz field in the $F_{x,\,in}$ waveform positioned at the time delay of 0 ps. The details of the time offset and the parameters used in our data fitting can be found in Supplementary Notes 4. We have also attempted to characterize the optical response of our devices under static, DC-bias conditions. However, all attempts to recreate the gating fields of the order of MV/cm in the gap region of our antenna by applying the necessary static voltage to the contact pads of the electrodes led to permanent damage of the devices via dielectric breakdown and unintentional shortcuts, as described in Supplementary Notes 5. This observation points to yet another advantage of using very fast

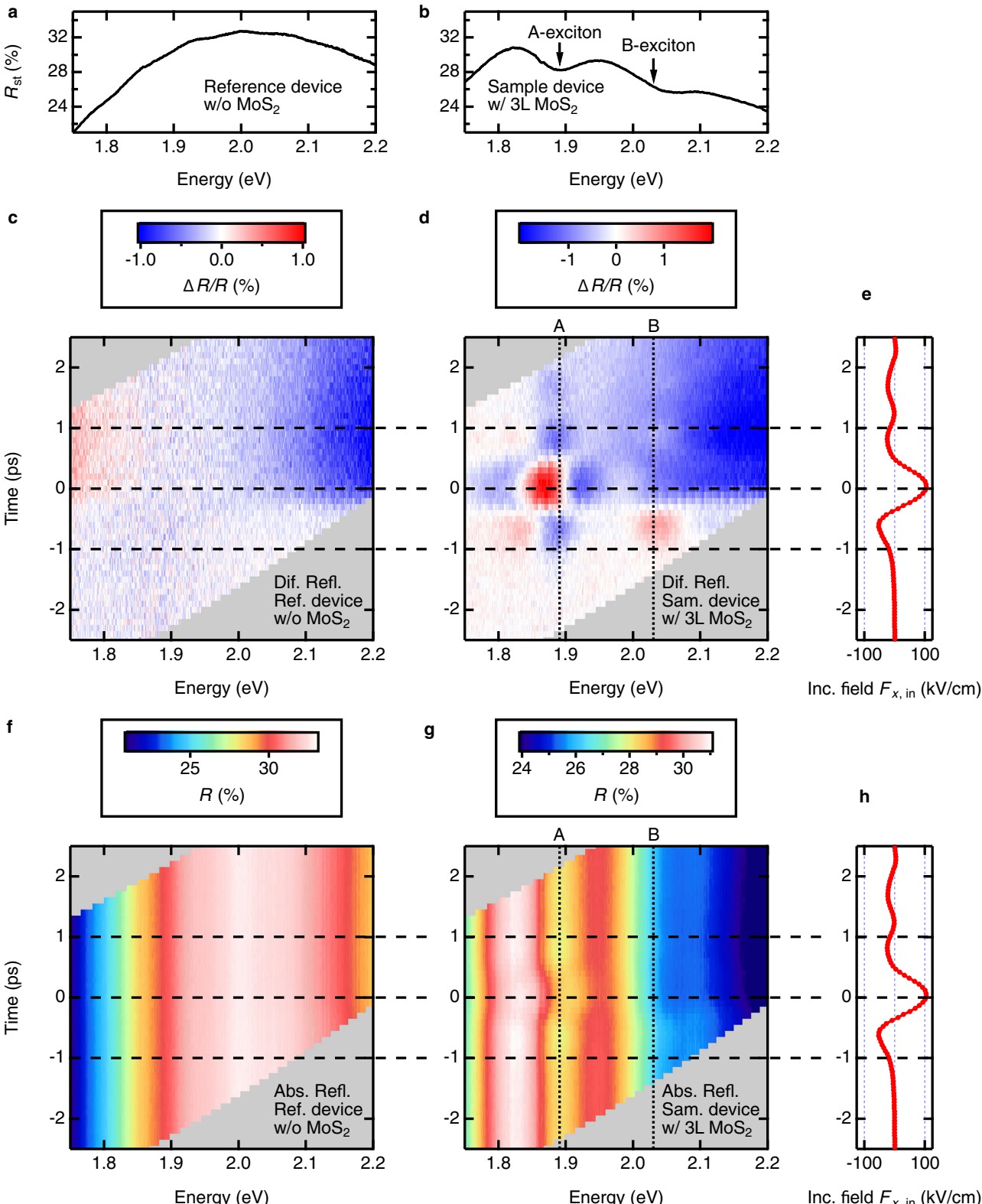

**Fig. 2 | THz pump-optical probe (TPOP) response of the devices from the production batch α.** Static absolute optical reflectance spectra $R_{st}$ of the antenna without MoS$_2$ (**a**) and with 3 L MoS$_2$ (**b**) measured in the absence of the THz field. A- and B-exciton resonances in MoS$_2$ are marked with arrows. Transient differential reflectance spectra $\Delta R/R$ (**c, d**) and absolute reflectance spectra $R$ (**f, g**), respectively, observed in a TPOP measurement with incident THz field $F_{x,in}$ shown in

(**e, h**). Here, (**c, f**) are the results of an antenna without MoS$_2$ and (**d, g**) are that of an antenna with 3 L MoS$_2$. Vertical dotted lines in (**d, e**) indicate the spectral positions of A- and B- excitons without THz pump. Horizontal dashed lines in (**c–h**) are guide-to-the-eye to provide the relative timing of a TPOP response in antennas with respect to the incident THz field.

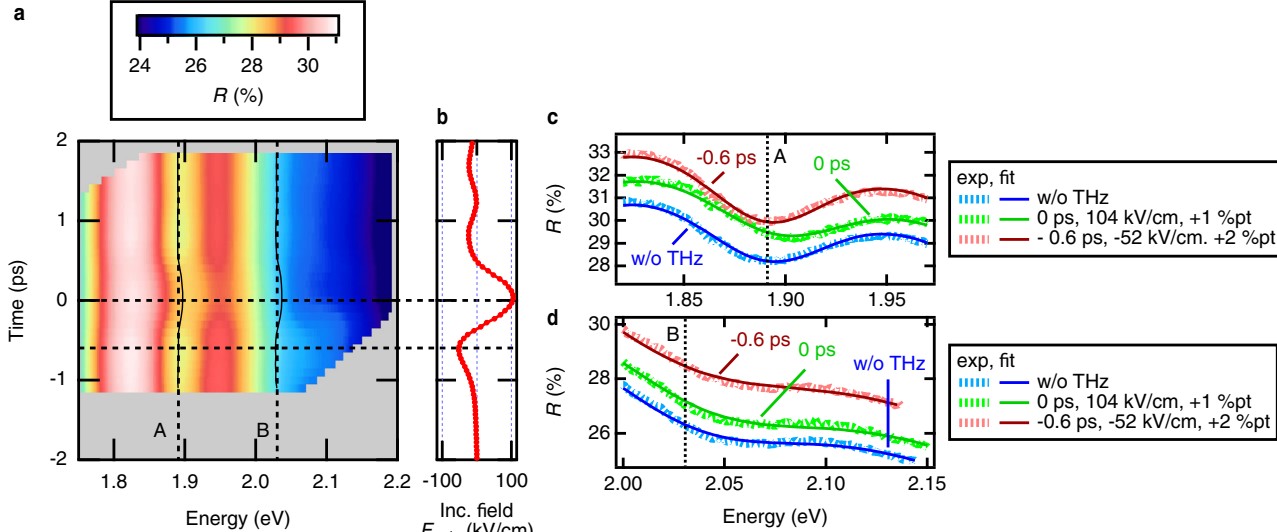

**Fig. 3 | The fitting of transient absolute reflectance spectra of a hybrid 3D-2D THz nanoantenna with 3 L MoS₂ from the production batch α. a** Transient reflectance spectra obtained via fitting of the experimental data. Two vertical dotted lines show the position of A- and B-exciton resonances without the presence of the THz field. Two solid curved lines show the temporal change of position of A- and B-exciton resonances. **b** Incident THz field $F_{x,\,in}$. Horizontal dotted lines in (**a**, **b**) point to time delays of 0 ps and −0.6 ps. Absolute reflectance spectra R in the vicinity of A-exciton (**c**) and B-exciton (**d**) resonances without THz pump (blue), under THz pump of $F_{x,\,in}$ = 104 kV/cm corresponding to 0 ps time delay (light green, vertically offset by +1 %pt for clarity), and under THz pump of $F_{x,\,in}$ = −52 kV/cm corresponding to −0.6 ps time delay (dark red, vertically offset by +2 %pt for clarity). Dotted lines are experimental data, and solid lines are fitting curves. The vertical dotted lines in (**c**, **d**) show the position of A- and B-exciton resonances, respectively, without the presence of the THz field.

oscillating THz fields for efficient gating of 2D materials as being non-destructive.

## Quantifying the effect of THz gating in MoS₂

In order to quantify the effect of the THz gating on the excitonic transitions in MoS₂ and to extract the THz field-dependent parameters of A- and B-exciton resonances, we have performed the fits of the time-dependent absolute reflectance spectra R of the THz-gated antenna shown in Fig. 2g. The reflectance spectra of the antenna with 3 L MoS₂ sample could be well described by a third-order polynomial describing the background convex curve, and the two Lorentzian dips representing A- and B-exciton resonances. The details of the fitting are described in "Methods" Section 5.

Figure 3a is an image plot of the fitted transient reflectance spectra, and Fig. 3b is the incident THz field $F_{x,\,in}$ measured at the position of the antenna. Measured and fitted optical reflectance spectra in the vicinity of A- and B-exciton resonances, corresponding to selected pump-probe delays are shown in Fig. 3c, d, respectively. These pump-probe delays of 0 ps and −0.6 ps correspond to the incident field strength of $F_{x,\,in}$ = 104 kV/cm and $F_{x,\,in}$ = −52 kV/cm, or to estimated gating fields of $F_z$ = 1.4 MV/cm and $F_z$ = −0.69 MV/cm, respectively. The dashed vertical lines in Fig. 3c, d mark the spectral positions of A- and B- exciton resonances without the THz field. Note that the presented optical reflectance spectra are distorted due to the presence of a spectrally broad convex background. This leads to an apparent shift of spectral minima with respect to the true spectral positions of the exciton resonances, which, however, can be reliably determined from the Lorentzian parts of the fit functions, as presented below. We note that in our experiments, the absolute polarity of the electric field within the incident THz waveform $F_{x,\,in}$ could not be determined. Therefore, in the context of our discussion, we define the electric field of the strongest half-cycle in $F_{x,\,in}$, presented in Figs. 1–4 as "positive".

In Fig. 4a, b, we present the dependence of the spectral positions of A- and B- exciton resonances, extracted from the fitting described above, on the THz pump-optical probe delay. The incident THz waveform $F_{x,\,in}$ is presented in Fig. 4c as a reference. One observes a clear spectral modulation of the transition energy of both A- and B-excitons, which is time-coherent with the dynamics of the incident THz field. The strongest effect is the blueshift of both resonances at the strongest "positive" half-cycle of the THz field oscillation, and both resonances also show a weaker redshift when the THz field changes its sign. This is the demonstration of the coherent nature of the THz gating field effect on the exciton resonances in MoS₂, realizing the ultrafast, sub-picosecond control of the fundamental properties of a 2D material via THz gating.

The maximum THz-driven modulation of resonant energy for both A- and B-exciton was of the order of 6-7 meV. We have established a dependence of the spectral shift of exciton resonances on the applied THz field, using the correlation between the field strength, including its sign, and the time delay within the incident THz waveform $F_{x,\,in}$. This correlation is illustrated by color coding in Fig. 4c–e. The THz-field dependence of the spectral positions of A- and B-excitons is presented in Fig. 4d, e, respectively. The strongly dominant blueshift at the "positive" THz gating field and much weaker redshift at the "negative" THz fields are observed for both exciton resonances. We have also tested the response of our devices to the THz gating field with the reversed polarity. For this, we rotated the antenna by 180° about the propagation axis of the incident THz beam (see Fig. 5a). The generally bipolar modulation of the exciton transition energies with applied gating THz fields of opposite signs and the observations related to the reversal of polarity of the THz gating field, which are detailed in the following section, point to an existence of permanent symmetry breaking in the z-direction in our structure comprising a hybrid 3D-2D nanoantenna and a MoS₂ flake. The possible origin of this symmetry breaking will be addressed below in the "Discussion" section.

## Performance variation among fabricated devices

As mentioned above, several devices with varying numbers of MoS₂ layers were fabricated and tested in this work. A total of two device batches were produced, termed here α- and β-batches. Besides the

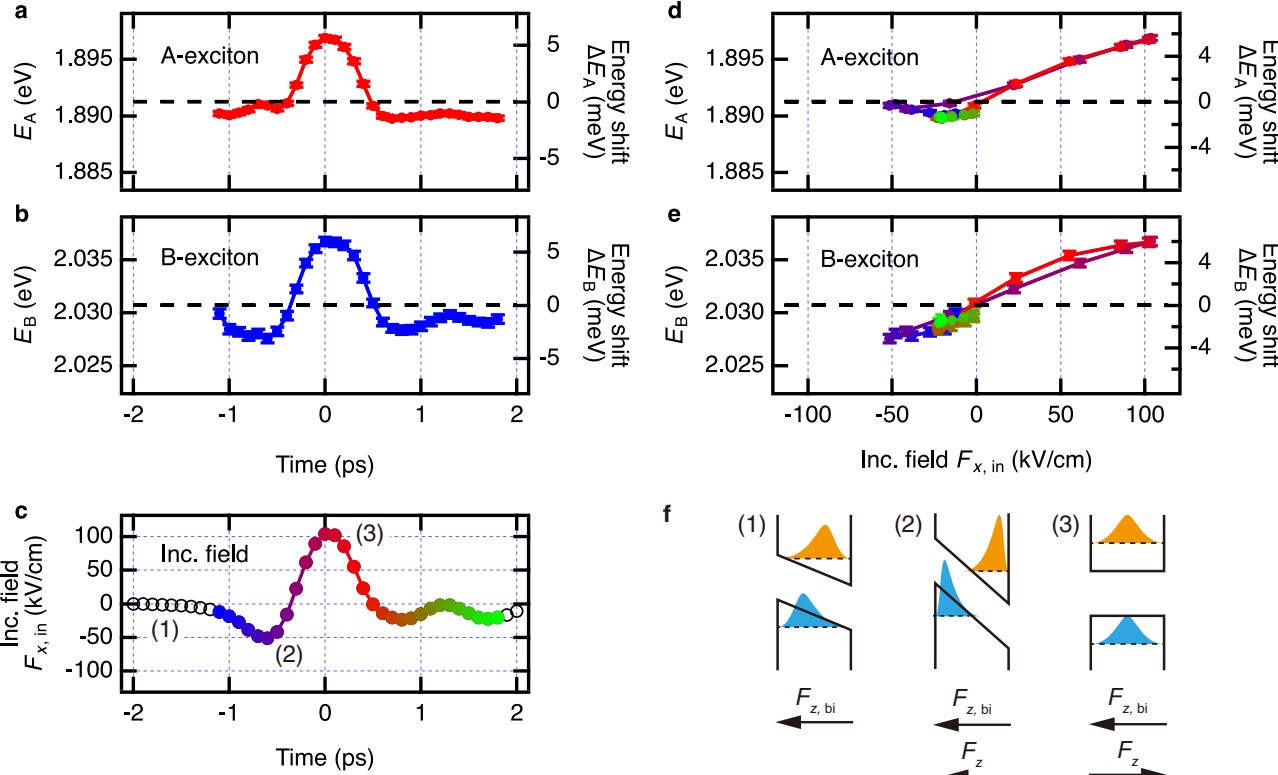

**Fig. 4 | Dependence of exciton resonance energy on THz pump-optical probe (TPOP) time delay and THz field strength for a device with 3 L MoS₂ from the production batch α. a, b** Dependence of the A-exciton resonance energy $E_A$ and B-exciton resonance energy $E_B$ on the TPOP time delay. $\Delta E_A$ and $\Delta E_B$ on the right axis are the shifts in the exciton resonance energies in the absence of the THz pump. Horizontal dashed lines show the resonance positions without the presence of the THz field. **c** Incident THz field $F_{x,\,in}$ in this measurement. Labels (1)-(3) indicate the timing corresponding to the schematic Figs. (1)-(3) in (**f**). **d, e** Dependence of the resonance energy for A- and B-excitons on the THz field strength. This dependence was established via the time correlation between the time-dependent

resonance energies in (**a, b**) and the time-dependent instantaneous THz field in (**c**). This time correlation is illustrated via color coding in (**c, d**, and **e**). Error bars in (**a, b, d**, and **e**) show the standard deviation of the exciton resonance energies as resulting from the corresponding fits. **f** Illustration of a quantum-confined Stark effect in a 2D semiconductor induced by an out-of-plane THz field $F_z$ under the presence of a built-in field $F_{z,\,bi}$. Sub-Figs. (1)–(3) correspond to the timing labels (1)–(3) in (**c**). The solid lines illustrate the quantum-confinement potential, the dashed lines depict the energy levels of the electron and hole, and the areas filled with orange and blue colors depict the squared wave functions of the electron and hole, respectively.

difference in the top electrode thickness (25 nm and 50 nm for the devices in α- and β- batches, respectively), another difference between the batches was the annealing condition during the fabrication, as described in Methods Section 1. Below, we present the variation in performance among the fabricated structures, crucial for the explanation of the symmetry breaking within the devices.

In Fig. 5a–f, the measured exciton spectral position shifts for three selected devices are shown: (I) A device with 3 L MoS₂ from α-batch, (II) a device with 3 L MoS₂ from β-batch, and (III) a device with 4 L MoS₂ from α-batch. These measurements include rotation of the antenna structure by 180° in the incident THz beam, thereby inverting the polarity of the THz gating field on MoS₂ (see Fig. 5a). Each individual measurement in Fig. 5 b–g is presented in a different color. For the measurement performed with the antenna rotated by 180°, the sign of the incident THz field in the horizontal axes in Fig. 5a–e was inverted. The results shown in Figs. 2–4 and discussed previously are related to the device (I). The individual measurement data for each device can be found in Supplementary Notes 6 and Supplementary Data 1.

Comparing the devices, we found that the bipolar THz gating can lead to a predominant blueshift (Fig. 5b, c), both blue- and redshifts (Fig. 5d, e), or a predominant redshift (Fig. 5f, g) of the exciton resonances, with both A- and B-exciton resonances following the same shift trend. The possible origin of this behavior is addressed in the Discussion section below.

## Discussion

As described above, we have demonstrated an efficient THz gating in MoS₂ embedded into a hybrid 3D-2D THz nanoantenna. The THz gating resulted in the pronounced shift of transition energy of A- and B-excitons in MoS₂, which is time-coherent with the incident broadband THz field. In addition, the polarity of the observed energy shifts varied among the studied devices and included predominant blueshift, bipolar shift, and predominant redshift of the exciton transition in response to the bipolar-oscillating THz gating field (see Fig. 5b–g).

In a 2D semiconductor without broken symmetry in out-of-plane direction, an application of an electric field of *any* polarity along the same direction will lead to an instantaneous redshift of a ground state optical transition—the well-known quantum-confined Stark effect (QCSE). In the perturbative regime, the magnitude of this Stark shift has a quadratic dependence on the electric field strength[44,45]. Let us assume that this electric field in a semiconductor pre-exists, and we refer to it as a built-in field. An application of an *additional* electric field along the same direction, such as, e.g., the THz gating field, depending on its polarity, will lead to either a weakening or to an enhancement of the pre-existing QCSE induced by a built-in field. This will, in turn, result in either the blueshift or the redshift of the optical transition energy, respectively. The direction and strength of the observed shift of the optical transition energy will thus depend on the polarity and strength of the pre-existing built-in field and that of the applied external gating field, as illustrated schematically in Fig. 4f.

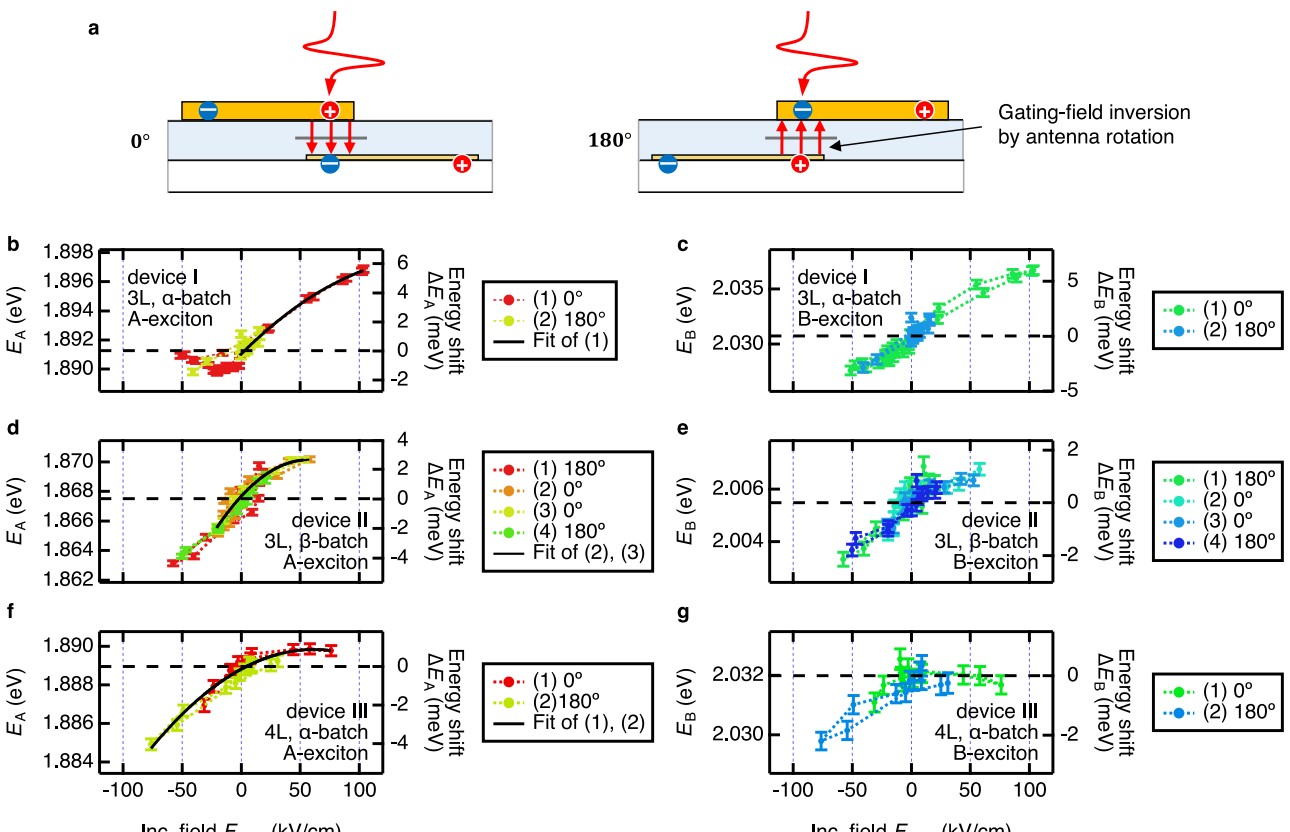

**Fig. 5 | Shift of exciton resonance energy in different devices and with different gating-field polarizations. a** Illustration of gating field inversion by rotating the antenna by 180° in the incident THz beam. The curved and straight red arrows represents the incident and gating THz field, respectively. Dependence of the A-exciton resonance energy $E_A$ (**b, d, f**) and B-exciton resonance energy $E_B$ (**c, e, g**) on the strength of the incident THz field $F_{x,in}$, for several tested devices: **b, c** – device I, the antenna with 3 L MoS₂ from production batch α, **d, e**—device II, the antenna with 3 L MoS₂ from production batch β, **f, g** device III, the antenna with 4 L MoS₂ from production batch α. These measurements included inversion of the gating field by rotation of antenna by 180°, as depicted in (**a**) and shown by the trace legends. Error bars in (**b**–**g**) show the standard deviation of the exciton resonance energies resulting from the corresponding fitting. Solid lines in (**b, d, f**): fit of the parabolic part of the dependence of A-exciton resonance energy on the incident THz field $F_{x,in}$ with a quadratic equation describing the perturbative-regime quadratic quantum confined Stark effect, and the polarizability of the A-exciton established in ref. 22. These fits allow one to estimate the strength of the built-in field in the tested devices, as well as the THz field enhancement factors of the antennas and the peak THz gating field strength.

The static field-induced QCSE is by now well studied in MoS₂[22–25] and other 2D semiconductors[26,27]. Here, we compare our THz gating-induced energy shifts of A- and B-exciton transitions with that from existing literature describing the DC field-induced Stark shifts in MoS₂. In ref. 22, Klein et al. studied the QCSE in mono- and few-layer MoS₂ films sandwiched between the two enclosing oxide layers, a geometry similar to the design of our hybrid 3D-2D nanoantenna (Fig. 1b). Excitonic Stark shifts of the order of 5-10 meV were observed in response to applied static gating fields of MV/cm scale, which is in line with our observations. It was also found in ref. 22 that the amount of the Stark shift in single- and multi-layer MoS₂ does not depend on the number of layers in the sample but only on the total applied field strength, which is also in agreement with our results. From this, it was concluded that the observed exciton transitions belong to intra-layer excitons in MoS₂. Further, it was found that several studied samples had an unintentional built-in electric field in the z-direction. The origin of this built-in field can be explained by the presence of the positively charged trap states near the MoS₂/oxide interface, such as border traps originating from oxygen vacancies within the oxide layer or interface traps originating from impurities/defects of MoS₂[46–48]. If the concentration of these positive traps is different for the two opposing MoS₂/oxide interfaces in the structure, e.g., due to intentional or unintentional difference in layer growth conditions, a potential difference, and hence a net built-in electric field in MoS₂ in the z-direction will be created, as was indeed

demonstrated in refs. 22,46–48. These uncompensated charged interface traps thus lead to the pre-existing QCSE in MoS₂/oxide structures in the z-direction, resulting in a generally bipolar shift of excitonic transition energy in response to a bipolar external gating field, which is in full agreement with our observations. For some samples in the literature, a deviation of the Stark shift from its initially quadratic field dependence, followed by its strong saturation with increasing gating field in one certain direction was also reported in ref. 22, again in agreement with our observations presented in Fig. 5b, d. In ref. 22, this saturation was explained by the charge transfer between MoS₂ and the charged trap states at the MoS₂/Al₂O₃ interface.

In a later study, Roch et al.[23] similarly observed the QCSE in a MoS₂ sample sandwiched between the two hexagonal boron nitride (hBN) layers and subject to MV/cm – scale DC bias fields. The hBN, in contrast to commonly used oxides such as Al₂O₃ or SiO₂, famously features a very low density of defects and charge trap states. Yet, the observed Stark shifts in ref. 23 were reported to be significantly smaller than those observed by Klein et al.[22] and by us, both using the MoS₂/oxide arrangement. This demonstrates the enhancement of exciton polarizability in a MoS₂/oxide arrangement as compared to the MoS₂/hBN one, once again highlighting the role of the environment in the physics of 2D materials[11–13].

Based on the above, we conclude that our observations—namely, the generally bipolar shift of exciton transition energy under bipolar

THz gating and the saturation of the quadratic QCSE at stronger gating fields—are consistent with the results of DC-Stark shift experiments presented in refs. 22,23. Our findings can be thus explained by the THz-induced QCSE in $MoS_2$ in the presence of a symmetry-breaking built-in electric field applied in the $z$-direction. This built-in field originates from an unintentional imbalance in the concentration of positive trap states at the opposing $MoS_2$/oxide interfaces within the structures, creating a potential difference between the two interfaces. Therefore, the observed variations in device performance can be attributed to different strengths of this built-in field, resulting from variation in interface trap concentrations among the devices. This is likely caused by the variation in fabrication conditions, most notably the device annealing process (see "Methods" Section 1).

With this physical picture in mind, we use the known dependence of the Stark shift on the applied static bias field in order to experimentally calibrate the THz gating field $F_z$ in our experiments. In ref. 22, using $MoS_2$/oxide arrangement and observing the Stark shift similar to ours, the A-exciton energy was described by the quadratic QCSE in the wide range of applied fields, and the polarizability of A-exciton was determined as $\mu = (0.58 \pm 0.25) \times 10^{-8} \, DmV^{-1}$. Using this value, we fit the parabolic parts of the observed field-dependent A-exciton transition energy $E_A(F_{x,\text{in}})$ from Fig. in 5b, d, f with the quadratic equation $E_A(F_{x,\text{in}}) = E_{A,0} - \mu(kF_{x,\text{in}} - F_{z,\text{bi}})^2$, where $E_{A,0}$ is the A-exciton transition energy in $MoS_2$ in the absence of QCSE, $F_{x,\text{in}}$ is the experimentally measured incident THz electric field, $k = |F_z/F_{x,\text{in}}|$ is the field enhancement factor of our antenna, and $F_{z,\text{bi}}$ is the pre-existing built-in field in the $MoS_2$ discussed above. For details, see Methods Section 6. The resulting fits are shown as black solid lines in Fig. 5b, d, f, yielding the following parameters. The exciton transition energy in the absence of QCSE was found to be $E_{A,0} = 1.8977 \pm 0.0008$ eV, $1.8701 \pm 0.0004$ eV, and $1.8898 \pm 0.0001$ eV for devices I, II, and III, respectively, which is in good agreement with refs. 9,10,22,23. The sample-to-sample variation in $E_{A,0}$ of about 15 meV is related to the difference in number of layers[9,10] and to dielectric disorder in the samples[12]. The built-in field in our antennas was found to be $F_{z,\text{bi}} = 2.4 \pm 0.1$ MV/cm, $1.5 \pm 0.1$ MV/cm, and $0.94 \pm 0.07$ MV/cm for devices I, II, and III, respectively, which is consistent with the values reported for similar $MoS_2$/oxide structures in ref. 22. We also determined the field enhancement factors of $k = 15 \pm 2$, $25 \pm 4$, and $15 \pm 1$ for devices I, II, and III, respectively. These experimental values are of the same order as our theoretical estimate of $k = |F_z/F_{x,\text{in}}| = 13.3$. While the experimental and the calculated field enhancement factors are in good agreement for devices I and III, the experimental value exceeds the theoretical prediction by almost a factor of two for device II. This scattering among the experimentally measured field enhancement factors is most likely caused by the variation of antenna fabrication conditions. The fact that our experimental values always exceed the theoretical ones can originate from the conservatively underestimated value of gold electrode conductivity used in our FDTD simulations, in turn leading to a smaller computed THz-driven current in the top and bottom electrodes and hence to a weaker conversion into the THz gating field than measured experimentally.

As already mentioned, in order to prevent potential damage to our devices, in our experiments, we kept the maximum gating field at a level not exceeding $|F_z| = 1.2$–$1.5$ MV/cm by monitoring the observed Stark shifts and using the field calibration as described above. For this, the incident THz field was attenuated in order not to exceed this maximum value of $|F_z|$ when necessary.

We note that apart from the THz-induced QCSE as described above, several alternative physical mechanisms could potentially also lead to the modification of optical absorption at the exciton resonances. However, none of these mechanisms would be fully compatible with our observations in the same way as the QCSE. Free-carrier generation and carrier heating in the $MoS_2$ via a strong THz field can be eliminated since these effects will be insensitive to the polarization of

the driving field, in contradiction to our observations shown in Fig. 5. THz-field-driven doping-level change in the $MoS_2$, which could occur if a $MoS_2$ flake were in good electrical contact with either electrode in the field-enhancement region, is also not likely. In this case, such an electrical spacer breakdown or current leak would have to occur in all our measured devices without exception. This is highly improbable, especially since all our devices feature well-insulating 75 nm-thick ALD-grown $Al_2O_3$ spacer layers.

In this work, we have demonstrated a pronounced THz field effect in a 2D semiconductor $MoS_2$ using a specially designed hybrid 3D-2D nanoantenna. It converts an incident broadband THz electromagnetic field into a vertical gating field within the material, simultaneously amplifying it by more than one order of magnitude to MV/cm level, while maintaining its full bandwidth of at least 0.1–2.5 THz. The THz field effect was observed via the time-resolved optical probing of characteristic exciton resonances in $MoS_2$, and the specific gating mechanism in this case was the quantum-confined Stark effect, which is time-coherent with the driving THz field.

Our approach to THz gating in 2D materials can be readily applied to numerous advances in nanotechnology and fundamental science. Besides the THz-driven QCSE described above, the ultrafast control of the doping level in 2D materials embedded in a hybrid 3D-2D nanoantenna is feasible. Here, the THz-injected carrier densities could become as high as $10^{13} \, cm^{-2}$ in case of direct contact between the 2D material and one of the electrodes (see Supplementary Notes 2–4), thus enabling f.ex., the direct THz-rate modulation of 2D transistors[2,15,16] via ultrafast control of the doping level in the channel. We also note that the carrier density of the order of $10^{13} \, cm^{-2}$ is sufficient even to induce a phase transition in a material such as $MoS_2$[16]. In case of direct contact between a 2D material and both electrodes, an ultrafast out-of-plane current in the 2D material can be induced, f.ex., enabling the ultrafast control of 2D memristors[7]. Other possible applications of our hybrid 3D-2D nanoantenna structures could be sub-picosecond activation of 2D photodetectors and optical modulators[4,17], memristor networks[49], direct THz control over the phase transitions including quantum phases in moiré systems[8,16,20,29], ultrafast manipulation of neutral and charged excitons in 2D semiconductors[28], direct electric field activation of THz layer-breathing phonons in van der Waals heterostructures[50,51], and many more.

Our THz gating method has significant potential for further development. The nanoantenna geometry and structure can be further optimized for more efficient resonant or non-resonant incoupling and conversion of the input control signals, also from other spectral ranges such as mid-infrared. It can also be further optimized to allow the non-destructive application of static bias fields of MV/cm scale, thereby realizing an active control over the polarity and depth of the THz field effect in 2D materials. The dielectric spacer layers enclosing the 2D material can be engineered in order to deterministically create the built-in bias fields within the structures via control of the interface trap states. Further, the electronic polarizability in a 2D material, and hence its basic response to the gating fields in the antenna, can be controlled via the suitable choice of dielectric layer materials such as high defect density $Al_2O_3$ or $SiO_2$, or low defect density hBN.

In summary, our results pave the way for technology and fundamental science of 2D materials where field effects induced on a sub-picosecond time scale are essential.

## Methods

### Device fabrication

The hybrid 2D-3D THz nanoantenna was fabricated as follows: (1) A 1.1-mm thick alkaline earth boro-aluminosilicate glass (Corning Eagle XG) substrate was prepared. (2) An 8-nm-thick Au layer was deposited via electron-beam lithography and sputtering to form the bottom electrode. (3) A 42-nm-thick Au layer was added for the connection line and contact pad of the bottom electrode via electron-beam lithography

and sputtering. (4) A 75-nm-thick $Al_2O_3$ layer was grown via atomic-layer deposition (ALD) as the bottom half of the spacer layer. (5) A few-layer $MoS_2$ flake was fabricated via mechanical exfoliation technique. (6) The $MoS_2$ flake was transferred above the square of the bottom electrode using the viscoelastic stamp technique[52]. (7) A 75-nm-thick $Al_2O_3$ layer was grown via ALD as the top half of the spacer layer. (8) The bottom contact pad was exposed by etching the top $Al_2O_3$ layer. (9) A 3-nm-thick Cr and a thick Au layer were deposited via electron-beam lithography and sputtering to form the top electrode. The thickness of the thick Au layer was 25 nm for $\alpha$-batch and 50 nm for $\beta$-batch. Samples of $\beta$-batch were annealed at 250 °C for 1 h in ambient conditions after step (9). The layer number of the $MoS_2$ flake was determined via Raman microscopy before step (6). The shape of the flake was monitored with an optical microscope to confirm that the desired layer number had been transferred to the target location.

## Device characterization
The leak current and dielectric-breakdown threshold for devices without $MoS_2$ flake were measured via a two-terminal method, as shown in Supplementary Notes 5. Although the samples from $\alpha$-batch had large sample-to-sample variation of leak current and breakdown threshold, these devices typically had a leak current of up to 8 to 40 $\mu$A under a DC voltage of 110–150 V and underwent dielectric breakdown at larger voltages. The samples from $\beta$-batch had much less variation, and had a leak current of 200 pA under a DC voltage of 20 V. The similar TPOP response observed for samples from $\alpha$- and $\beta$-batch regardless of the large difference of the insulation of the leak current indicates that the TPOP response is not caused by the leak current.

Photoluminescence (PL) spectra of the $MoS_2$ at the antenna gap were measured via PL microscopy. For each device, a PL peak corresponding to A-exciton luminescence was observed. This result supports our assignment of the A-dip to the A-exciton transition (for detail, see Supplementary Notes 7).

## Optical measurements
We used femtosecond laser from a Ti;Sapphire regenerative amplifier (Solstice, Spectraphysics) with center wavelength of 800 nm, spectral bandwidth of 50 nm (FWHM), repetition rate of 1 kHz, pulse energy of 6 mJ for the reflection microspectroscopy, TPOP reflection microspectroscopy, and electro-optic (EO) sampling measurement. The laser was divided into THz-generation and optical-probe pulse via a beam splitter. The probe pulse attenuated to approximately 1 mW was loosely focused on a 4-mm-thick sapphire crystal to generate a white-light pulse as a probe of the microspectroscopy covering the range of 1.75–2.2 eV. The white light was split into sample and reference beam using a cube beam splitter to obtain the reflectance spectra. An imaging spectrometer (Shamrock 303i, Andor) with a grating of 150 l/mm and a CCD camera (iKon-M 934, Andor) was used to simultaneously measure the spectra of the sample and reference beam focused in the entrance slit of spectrometer. The spectral resolution was characterized as 0.8 nm (FWHM) by measuring a spectrum of a HeNe laser.

For the reflection microspectroscopy, the sample beam attenuated to 100 fJ/pulse was focused on the field-enhancement region of the antenna from the backside of the substrate, using a 20x objective lens (LCD Plan Apo NIR 20x (t1.1), Mitsutoyo.). The size of the focused beam was approximately 2 $\mu$m as radius of the dark ring of Airy disk, corresponding to peak fluence of 3 $\mu$J/cm$^2$. The spectra of the reflected sample beam and reference beam were measured simultaneously. A reflection spectrum of protected silver mirror (P01, Thorlabs) was measured to calibrate the spectral transfer function of the setup and to derive the absolute static reflectance spectra $R_{st}$ of the devices.

For the TPOP measurement, the THz-generation pulse of approximately 2 W was used for THz-pulse generation in a LiNbO$_3$ with a tilted-pulsefront scheme[41,42]. The THz pulse was focused on the

antenna device from the front side of the substrate. The sample beam attenuated to 900 fJ/pulse was focused on the field-enhancement region using a 10x objective lens. The radius of dark ring of Airy disk was approximately 5 $\mu$m, corresponding to the fluence of 4 $\mu$J/cm$^2$. By measuring the spectra of the reflected sample beam and reference beam while chopping the THz pump, transient differential reflectance spectra $\Delta R/R$ under the THz pump was obtained. The time delay between the pump and probe was swept to obtain the transient differential reflectance. The amplitude of THz pulse was attenuated with high resistivity silicon wafers to approximately 100 kV/cm to avoid the dielectric breakdown of the devices which was observed in the higher field. The transient absolute reflectance spectra were obtained from $R_{st}$ and $\Delta R/R$ as $R = R_{st}(1 + \Delta R/R)$. We note that the out-of-plane THz field in the field-enhancement region has slight spatial inhomogeneity, and the observed spectral change is the average of the effect of the inhomogeneous field. However, the inhomogeneity is small enough and does not affect our observation in any significant way (see Supplementary Notes 1–6).

An EO sampling measurement[42] at the sample position in a transmission geometry with a 1-mm thick ZnTe was performed to measure the THz-field waveform. For more detail on the measurements with the setup geometry, see Supplementary Notes 8.

## De-chirping of the optical probe pulse
The chirp of the optical probe pulse was separately characterized, and the numerical chirp correction was applied in the data processing in order to ensure the maximum time resolution in our measurements of ca 100 fs. We note that due to a very low pulse fluence of the order of 1 $\mu$J/cm$^2$, the probe field interaction with the antenna structure and the 2D material remains linear for both chirped and compressed probe pulses[53] with and without the THz field present. Therefore, numerical chirp correction of the probe pulse, performed at the stage of the data processing, cannot influence the dynamics of the presented results. See Supplementary Notes 3 for more detail.

## Fitting of transient reflectance spectra
In the fitting of transient absolute reflectance spectra, we fitted the reflectance spectra at each TPOP time delay with the following equation: $R(E, t) = f(E) - Lor_A(E, t) - Lor_B(E, t)$, where $f(E)$ is the third-order polynomial which does not vary in time, and $Lor_A(E, t)$ and $Lor_B(E, t)$ are Lorentzian shape corresponding to A- and B-dips. Amplitudes, widths, and center energies of $Lor_A(E, t)$ and $Lor_B(E, t)$ are fitting parameters at each time delay. Global fitting was used to fit all the spectra with the common time-constant background $f(E)$ and the time-dependent Lorentzian parameters.

## Fitting of exciton transition energy
In the fitting of exciton transition energy against incident THz field in Fig. 5, to fit the data point without significant saturation in the lower-energy side while maximizing the available data points, we fitted in the following way: For Fig. 5a, we used the trace (1) and limited the fitting range to the range without significant saturation in the lower-energy side. In the fitting of Fig. 5c, e, we used multiple traces without significant saturation on the lower-energy side without limiting the range and performed a global fitting by linking all the fitting parameters. The obtained fitting parameters are as follows: From the fitting of trace (1) in Fig. 5b, $E_{A, init} = 1.8977 \pm 0.0008$ eV, $F_{z, bi} = 2.4 \pm 0.1$ MV/cm, $k = 15 \pm 2$, from the global-fitting of traces (2) and (3) in Fig. 5d, $E_{A, init} = 1.8701 \pm 0.0004$ eV, $F_{z, bi} = 1.5 \pm 0.1$ MV/cm, $k = 25 \pm 4$, and from the global-fitting of traces (1)-(3) in Fig. 5f, $E_{A, init} = 1.8898 \pm 0.0001$ eV, $F_{z, bi} = 0.94 \pm 0.07$ MV/cm, $k = 15 \pm 1$.

## Reporting summary
Further information on research design is available in the Nature Portfolio Reporting Summary linked to this article.

## Data availability

The Source Data underlying the figures of this study are available with the paper. All raw data generated during the current study are available from the corresponding authors upon request. Source data are provided with this paper.

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

## Acknowledgements

We acknowledge the financial support from the European Union's Horizon Europe (HORIZON-MSCA-2021-PF-01, Project number 101060427—UCoCo), Horizon 2020 research and innovation program (EXTREME-IR—Grant Agreement No. 964735), Deutsche Forschungsgemeinschaft (DFG) within Project No. 468501411-SPP2314 INTEGRATECH and Project No. 518575758 HIGHSPINTERA, Bundesministerium für Bildung und Forschung (BMBF) within Project No. 05K2022 PBA Tera-EXPOSE, and Bielefelder Nachwuchsfond. We are grateful to Dr. Mohammed Nouh, Dr. Arslan Usman, Dr. Hilary Masenda, and Prof. Martin Koch for useful instructions regarding the 2D materials fabrication. We are also grateful to Dr. Shinya Takahashi, Dr. Kohei Nagai, Dr. Satoshi Kusaba, Dr. Kento Uchida, Prof. Takashi Arikawa, and Prof. Koichiro Tanaka for fruitful discussions throughout the entire project, and assisting with the characterization of the test devices using their real-time THz near-field microscope. We are grateful to Prof. Heejae Kim, Prof. Ryusuke Matsunaga, Prof. Ikufumi Katayama, and members of Ultrafast Science research unit at Bielefeld University for fruitful discussions and support throughout this project.

## Author contributions

T.H. and D.T. conceived and coordinated the project. T.H., S.N., S.F., H.S., and A.T. designed, fabricated and performed basic characterization of the hybrid 2D-3D THz nanoantenna. T.H., W.Z. and H.H. designed and built the measurement infrastructure. T.H. conducted the optical measurements and analyzed the data together with D.T. S.R. performed optical simulations. S.F., H.S., A.T., and D.T. supervised the project. T.H. and D.T. wrote the paper. All co-authors discussed the results and commented on the manuscript.

## Funding

## Competing interests

The authors declare no competing interests.
