## [Transparent Peer Review file · Nature Communications]

Terahertz field effect in a two-dimensional semiconductor

Corresponding Author: Professor Dmitry Turchinovich

Version 0:

Reviewer comments:

Reviewer #1

(Remarks to the Author)

Dear Editor,

T. Hiroaka and coworkers submitted their work entitled “Terahertz field effect in a two-dimensional semiconductor MoS₂” for consideration to Nature Communications.

I appreciate their work and find their approach smart and innovative. The claim is solidly supported by the results, which are presented in a very clear way. The fundamental photophysical effects observed due to the combination of intense THz field and the design of the nanoantenna are a breakthrough in the 2D materials and 2D materials-based device characterization fields. The scientific methodology is robust, as well as the data analysis.

The weak part of this manuscript is the lack of a target application where the THz-induced field effect can be exploited. This would broaden the audience of the work to the scientific community working on the integration of 2D materials in practical applications. I suggest the author, as a revision of the manuscript, to envision a technological application starting from the following paper and the literature therein:

-) Fiori, G., Bonaccorso, F., Iannaccone, G. et al. Electronics based on two-dimensional materials. *Nature Nanotech* 9, 768–779 (2014)

-) R. Ge et al, Atomrstor: Nonvolatile Resistance Switching in Atomic Sheets of Transition Metal Dichalcogenides, *Nano Lett.* 2018, 18, 1, 434–441

After this minor revision, I suggest the publication of this manuscript in Nature Communications

Reviewer #2

(Remarks to the Author)

This paper reports on a first demonstration of the use of single cycle THz fields to apply strong out-of-plane biases to a 2d material on picosecond time-scales. The observed modulation to the optical properties is shown to be associated with an effective Stark shift of exciton resonances in MoS₂. These are subtle effects and not too surprising (shown to be roughly consistent to that observed under DC fields). However the impact of this work is in the development of an apparatus which can translate in-plane THz fields into much stronger out-of-plane fields, thus enabling an all-optical bias which can be broadly applied to many different material systems in a capacitor like geometry. I think this is likely to have a significant impact and there are likely other material systems where such a setup can induce much more interesting field-driven responses. I am very much in support therefore of the publication of this work, which opens up a new means of driving materials and devices with strong electric fields (all prior work to my knowledge is focused on in-plane excitation).

The paper is very well written and easy to read and I only have one minor suggestion re future work: As noted prior static biases in this geometry have been achieved via electrostatic gating which allows for not only the application of applied fields but also doping of material systems. If it were possible to induce changes in the doping of materials on such fast time scales I think this would add an additional level of interest to the work. Is this a possibility or is the capacitance of the device a limiting factor here which would limit the magnitude of possible doping modulations on fast time-scales?

Reviewer #3

(Remarks to the Author)

This manuscript presents an innovative approach to ultrafast gating of MoS₂ using a terahertz (THz) field, enabled by a hybrid 3D-2D antenna structure. While the THz-induced Stark effect has been explored in quantum dots and the DC Stark effect observed in 2D materials, this work uniquely converts an in-plane THz field into a vertical field via a specially designed 3D antenna, allowing direct observation of the THz-induced Stark effect in MoS₂. These findings could contribute to the development of ultrafast 2D modulators.

However, several aspects of the manuscript could be improved, particularly in clarifying explanations, considering alternative mechanisms, and refining methodological details. Below are specific suggestions:

1. The manuscript attributes the THz-induced exciton shift to the Stark effect. Could other factors, such as free carrier generation or hot carrier effects, also play a role? A discussion of alternative explanations, along with comparisons to other THz-induced excitonic effects (e.g., *Adv. Opt. Mater.* 10, 2102407; *Nano Lett.* 20, 5214), would strengthen the interpretation.
2. Photoluminescence (PL) spectra are included in the supplementary file but are not discussed in detail. Could they provide insights into possible charge transfer effects induced by the THz field? While such effects typically persist longer than the observed response, luminescence measurements could help verify this.
3. The nanoantenna design is well described, but how does spatial inhomogeneity in field enhancement affect the excitonic response? A discussion on potential non-uniform field distributions would be valuable.
4. The manuscript discusses the damage threshold under a DC field but does not observe breakdown under THz pulses. Could increasing the THz field induce phase transitions or other nonlinear effects (*Nat. Commun.* 14, 5905)? Comparing the THz-induced field strength with the breakdown limits of Al₂O₃ and MoS₂ would help confirm whether dielectric damage occurs under ultrafast gating conditions.
5. The THz-induced shift is measured indirectly via optical reflectance. Have the authors considered direct electro-optic sampling to probe the local THz field in MoS₂?
6. Given that THz fields oscillate on a sub-picosecond timescale, does the optical probe's time resolution sufficiently capture the fastest THz-induced changes?
7. Some technical terms require clearer definitions for interdisciplinary readers. For instance, a simple energy level diagram illustrating the "THz-induced Stark shift" would aid understanding.
8. The conclusion could better highlight potential applications, such as ultrafast optical modulators or THz-driven excitonic switches.
9. The supplementary section would benefit from improved clarity. For example, the description "Figure F1. Template named as ***_summary_measTPOPdata, which shows the measured TPOP response of each antenna. *** is an identifier for each experiment." could be rewritten for better readability.

Version 1:

Reviewer comments:

Reviewer #3

(Remarks to the Author)

After the revision, I recommend this work for publication.

Dear Editor, dear Reviewers,

Thank you very much for working on our manuscript. We greatly appreciate the Reviewers' positive evaluation of our work, their valuable input regarding its further improvement, and their recommendation to publish it in Nature Communications after the revision.

The manuscript is now revised according to the Reviewers' comments, which we address below point by point. The Reviewers' comments are typed in black, whereas our response is typed in blue. The corresponding changes implemented in the manuscript and supplementary material are highlighted in yellow in their marked-up versions.

We hope that we addressed all the Reviewers' comments in a satisfactory manner, and that our manuscript can now be accepted for publication.

Best regards,

Tomoki Hiraoka and Dmitry Turchinovich, on behalf of all co-authors

Reviewer 1

T. Hiraoka and coworkers submitted their work entitled "Terahertz field effect in a two-dimensional semiconductor MoS₂" for consideration to Nature Communications.

I appreciate their work and find their approach smart and innovative. The claim is solidly supported by the results, which are presented in a very clear way. The fundamental photophysical effects observed due to the combination of intense THz field and the design of the nanoantenna are a breakthrough in the 2D materials and 2D materials-based device characterization fields. The scientific methodology is robust, as well as the data analysis.

We greatly appreciate the Reviewer's thorough evaluation of our manuscript, and his/her very favorable opinion regarding the quality, novelty and impact of our work. We are very grateful to the Reviewer for suggesting the improvements to our manuscript.

The weak part of this manuscript is the lack of a target application where the THz-induced field effect can be exploited. This would broaden the audience of the work to the scientific community working on the integration of 2D materials in practical applications. I suggest the author, as a revision of the manuscript, to envision a technological application starting from the following paper and the literature therein:

-) Fiori, G., Bonaccorso, F., Iannaccone, G. et al. Electronics based on two-dimensional materials. Nature Nanotech 9, 768–779 (2014)

-) R. Ge et al, Atomristor: Nonvolatile Resistance Switching in Atomic Sheets of Transition Metal Dichalcogenides, Nano Lett. 2018, 18, 1, 434–441

After this minor revision, I suggest the publication of this manuscript in Nature Communications.

We agree with the Reviewer that extending the discussion of possible target applications of THz field effect in 2D materials will indeed make our paper stronger, and we appreciate the Reviewer's suggestions regarding cited references. The suggested papers are dealing with the applications of 2D materials in CMOS and memristor technology, and the THz-induced field effect should be definitely a promising way to improve these technologies.

We have now cited the suggested papers in the Introduction describing the current technologies based on 2D materials. Further, in Conclusion, we also cited these papers while discussing various possible 2D-3D antenna configurations for different working mechanisms, i.e., applying an out-of-plane electric field, controlling the doping level, and inducing the out-of-plane current. We have also estimated the possible doping-level change achievable with our scheme, which is now presented its detail in the newly added Supplementary Section B4.

Reviewer 2

This paper reports on a first demonstration of the use of single cycle THz fields to apply strong out-of-plane biases to a 2d material on picosecond timescales. The observed modulation to the optical properties is shown to be associated with an effective Stark shift of exciton resonances in MoS₂. These are subtle effects and not too surprising (shown to be roughly consistent to that observed under DC fields). However, the impact of this work is in the development of an apparatus which can translate in-plane THz fields into much stronger out-of-plane fields, thus enabling an all-optical bias which can be broadly applied to many different material systems in a capacitor like geometry. I think this is likely to have a significant impact and there are likely other material systems where such a setup can induce much more interesting field-driven responses. I am very much in support therefore of the publication of this work, which opens up a new means of driving materials and devices with strong electric fields (all prior work to my knowledge is focused on in-plane excitation).

We greatly appreciate the Reviewer's thorough evaluation of our manuscript, and his/her very favorable opinion regarding the quality, novelty and impact of our work. We are very grateful to the Reviewer for suggesting the improvements to our manuscript.

The paper is very well written and easy to read and I only have one minor suggestion re future work: As noted prior static biases in this geometry have been achieved via electrostatic gating which allows for not only the application of applied fields but also doping of material systems. If it were possible to induce changes in the doping of materials on such fast time scales I think this would add an additional level of interest to the work. Is this a possibility or is the capacitance of the device a limiting factor here which would limit the magnitude of possible doping modulations on fast time-scales?

We are glad that the Reviewer is satisfied with the quality of our writing, and are grateful for his/her comment regarding the possible ultrafast doping level control in 2D materials using our scheme, which we address below. Indeed, it should be able to make a similar antenna structure for this purpose by omitting one of the spacer layers (top or bottom) to enable a direct contact between the 2D material flake and an electrode. In this case, the surface carrier density injected into a 2D material should be of the same order as the THz-induced surface carrier density change in the electrode in the antenna of current design. This is estimated at ca. $2 \times 10^{12}/\text{cm}^2$, as described in the newly added Supplementary Section B4. With some optimization of the structure, such as reducing the capacitance, it should be possible to obtain the doping change of ca. $10^{13}/\text{cm}^2$, which is high enough to induce a significant change of material properties in 2D materials [B. Radisavljevic, A. Kis, Nat. Mater. 12, 815–820 (2013).][K. F. Mak et al., Nat. Mater. 12, 207–211 (2013).]. Therefore, such a device should be a promising platform for the material control via ultrafast doping modulation. We now also emphasized this point in the Conclusion, while discussing the possible future applications of our technology.

Response to Reviewer 3

This manuscript presents an innovative approach to ultrafast gating of MoS₂ using a terahertz (THz) field, enabled by a hybrid 3D-2D antenna structure. While the THz-induced Stark effect has been explored in quantum dots and the DC Stark effect observed in 2D materials, this work uniquely converts an in-plane THz field into a vertical field via a specially designed 3D antenna, allowing direct observation of the THz-induced Stark effect in MoS₂. These findings could contribute to the development of ultrafast 2D modulators.

However, several aspects of the manuscript could be improved, particularly in clarifying explanations, considering alternative mechanisms, and refining methodological details. Below are specific suggestions:

We are very grateful to the Reviewer for his/her positive evaluation of our results, careful reading of our manuscript, and suggesting many points that deepen our discussion.

1. The manuscript attributes the THz-induced exciton shift to the Stark effect. Could other factors, such as free carrier generation or hot carrier effects, also play a role? A discussion of alternative explanations, along with comparisons to other THz-induced excitonic effects (e.g., Adv. Opt. Mater. 10, 2102407; Nano Lett. 20, 5214), would strengthen the interpretation.

We appreciate this valuable comment that allowed us to deepen our discussion. As the Reviewer suggests, below we will discuss possible alternative mechanisms such as (1) free carrier generation via a strong THz field, (2) carrier heating via a strong THz field, (3) Franz-Keldysh effect (FKE) [Nano Lett. 20, 5214], and (4) change of transition selection rules via QCSE [Adv. Opt. Mater. 10, 2102407]. In addition, we also discuss the (5) change in the doping level of MoS₂ as a possible mechanism. Based on these discussions, we conclude that our explanation, the absorption peak shift via QCSE, remains the only physical mechanism fully compatible with all the observations done in this work.

(1, 2) Regarding the possible free carrier generation and carrier heating in vertical THz fields:

In all our experiments, we observe the response of the MoS₂, which is *strongly sensitive to the polarity* of the applied THz field. This is fully compatible with the physical picture of the quantum-confined Stark effect (QCSE) induced by the vertical THz field in a MoS₂ flake subject to a strong (1-2 MV/cm) static bias built-in field. On the other hand, the effects such as free carrier generation and carrier heating via strong THz field *would be insensitive to the polarity* of the applied THz field. Therefore, we can exclude them from consideration.

Further, the additionally-generated free carriers would screen the symmetry-breaking built-in field in our devices, thus reducing the pre-existing QCSE, and hence resulting in the blue shift of the exciton resonances *persistent on the time-scale of the free-carrier lifetime*. The free-carrier recombination lifetime for few-layer MoS₂ is of the order of 100 ps [H. Wang, C. Zhang, F. Rana, Nano Lett. 15, 8204–8210 (2015)]. However, all the observed dynamics in our experiment is on the (sub-)ps timescale, fully compatible with the quasi-instantaneous QCSE but not with the picture of long-living free carriers generated by the strong THz fields.

(3) Regarding the possible Franz-Keldysh effect (FKE) in vertical THz fields:

FKE occurs when a strong electric field is applied in the direction in which the material has translational symmetry, as seen in the previous study of TPOP on MoS₂ with an in-plane THz field [Nano Lett. 20, 5214]. However, in our case, the sample is a few-layer MoS₂ (possibly the interface trap states are involved, too), in which the exciton is confined in the out-of-plane direction. Therefore, the FKE picture is not applicable in our case, and the QCSE is the only natural picture.

(4) Regarding the possible change of transition selection rule via QCSE

In the reference [Adv. Opt. Mater. 10, 2102407] the modulation of optical absorption in quantum dots (QD) was observed, mainly caused by the change of transition dipole moment via QCSE. The QD sample has IR-active and IR-forbidden excitonic transitions, and the external field significantly alternates the transition dipole moment and selection rule in this case. This effect, however, will strongly depend on the wavefunction symmetry and the degree of confinement in the sample.

We note here, that in the previous experiments studying the QCSE in MoS₂ using the MV/cm-level dc bias fields, *no significant change of the transition dipole moment* in the excitonic transitions was observed [Klein, J. *et al.* Stark Effect Spectroscopy of Mono- and Few-Layer MoS₂. *Nano Lett.* **16**, 1554–1559 (2016).][Roch, J. G. *et al.* Quantum-Confined Stark Effect in a MoS₂ Monolayer van der Waals Heterostructure. *Nano Lett.* **18**, 1070–1074 (2018).]. Otherwise, the observations made at dc are fully consistent with our THz results. Therefore, we also eliminate the change of transition dipole moment of excitonic species as an explanation of our results.

(5) Regarding the possible change of doping level in MoS₂ in vertical THz fields

As already described in our response to the Reviewer 2, the carrier density change on the electrode's surface in the gap region is estimated as ca. $2 \times 10^{-12} \text{ cm}^{-2}$. Therefore, if the electronic insulation of one of the spacer layers (top or bottom) is broken and the MoS₂ flake is in good contact with only one of the electrodes, the doping level in MoS₂ can be modulated by ca. $2 \times 10^{-12} \text{ cm}^{-2}$ on the ultrafast time scale. Such a change in doping level in MoS₂ will affect the absorption peak ratio of exciton and trion similarly to the already-known photoluminescence intensity change of exciton and trion [K. F. Mak *et al.*, *Nat. Mater.* **12**, 207–211 (2013)]. Also, the sign of the doping-level change would depend on the polarity of the incident THz field. Therefore, such an ultrafast doping-level change could, in principle, be the candidate for the ultrafast absorption peak shift mechanism.

However, our devices have the top and bottom spacer layers made of 75-nm-thick ALD-grown Al₂O₃. It is rather difficult to imagine a situation in which *all the devices observed in this study* would experience the same way of spacer breakdown or current leak through the spacer layer. Therefore, we would eliminate the unintentional ultrafast doping-level change as the mechanism to explain the observed spectral change in our samples.

We have included a paragraph summarizing the above consideration in the last part of the Discussion section. Again, we are very thankful to the Reviewer for giving us this opportunity to deepen our discussion.

2. Photoluminescence (PL) spectra are included in the supplementary file but are not discussed in detail. Could they provide insights into possible charge transfer effects induced by the THz field? While such effects typically persist longer than the observed response, luminescence measurements could help verify this.

We thank the Reviewer for suggesting another perspective for the physics in our system that is not covered in our study. Unfortunately, our PL measurement is not time-resolved, and therefore, it is difficult to discuss the effect of the THz pump via PL. It requires another experimental setup for time-resolved PL, e.g., a setup with a streak camera as a spectrometer detector, and it is beyond the scope of this study. Measuring the time-resolved PL with a THz pump would be indeed an interesting future task, since it could clarify the long-living effects, such as doping-level change.

3. The nanoantenna design is well described, but how does spatial inhomogeneity in field enhancement affect the excitonic response? A discussion on potential non-uniform field distributions would be valuable.

We are very grateful to the Reviewer for this comment, leading to an additional investigation on our side, which is now included into the revised manuscript.

Our antenna generates the out-of-plane field in the gap with spatial inhomogeneity that is unique to this system: Due to the spatial charge distribution defined by the antenna's resonance, the out-of-plane field in the field-enhancement region has a *small inhomogeneity in the x-direction*, as shown in the simulation Fig. R1.

Figure R2 shows the simulated waveforms of the THz field probed at three different positions in the antenna gap. The waveforms have a delay of max. 160 fs (much smaller than the THz cycle duration of ca 1.5 ps) and a peak amplitude difference of max. 12 % for the two points located on the opposite side of the gap in the x-direction, i.e., under the condition of maximum field inhomogeneity.

In our measurement, the effect of such a spatially distributed THz field is averaged by the size of the optical probe spread over the gap region. However, as we see in Fig. R2a and b, the maximum expected difference in the probed waveforms is very minor, not affecting the observed result in any significant way.

In addition, in relation to the spatial inhomogeneity of the charge distribution, we have also checked the amplitude of the in-plane THz field in the gap. As shown in Fig. R3, the in-plane component is smaller than the out-of-plane component by two orders of magnitude. Therefore, the effect of the inhomogeneity on the in-plane field component would be fully negligible.

We have therefore referred to the spatial inhomogeneity in Methods Section 3, and described the discussion above in a newly added Supplementary Section A6.

Figure R1. Spatial distribution of simulated out-of-plane THz field around the field-enhancement region at $z = 0$ plane plotted for various timing around the main peak of the THz pulse. The time for each frame is **a**: 6.5 ps, **b**: 7.0 ps, **c**: 7.5 ps, and **d**: 8.0 ps. The field-enhancement region of $10 \mu\text{m} \times 10 \mu\text{m}$ is shown by the solid square in each frame.

Figure R2. Simulated out-of-plane THz field waveform in the field-enhancement region probed at different points. Coordinate of the probed points are $(x,y,z) = (-4 \mu\text{m}, 0 \mu\text{m}, 0 \mu\text{m}), (0 \mu\text{m}, 0 \mu\text{m}, 0 \mu\text{m}), (4 \mu\text{m}, 0 \mu\text{m}, 0 \mu\text{m})$, respectively. **a**, Temporal waveforms plotted over 0-28 ps. **b**, Temporal waveforms plotted over 4-12 ps. The boxes labelled as c and d indicate the region plotted in **b** and **c**. **c**, Magnified view of the waveforms around their zero-crossing points around 8 ps. **d**, Magnified view of the waveforms around their peaks around 7.5 ps.

Figure R3. Comparison of the out-of-plane and in-plane THz field in the antenna gap probed at $(x,y,z) = (0 \mu\text{m}, 0 \mu\text{m}, 0 \mu\text{m})$. The in-plane component is smaller than the out-of-plane component by two orders of magnitude.

4. The manuscript discusses the damage threshold under a DC field but does not observe breakdown under THz pulses. Could increasing the THz field induce phase transitions or other nonlinear effects (Nat. Commun. 14, 5905)? Comparing the THz-induced field strength with the

breakdown limits of Al₂O₃ and MoS₂ would help confirm whether dielectric damage occurs under ultrafast gating conditions.

We thank the Reviewer for pointing out the missing information in our manuscript. Actually, we did observe a breakdown in some preliminary test samples before batch α in a TPOP experiment with incident THz field strength exceeding 100 kV/cm. However, we could not quantify the damage threshold of the device since the dielectric breakdown is a stochastic event and is sensitive to the difference in minor details of sample quality and experimental conditions. Also, we could not sacrifice too many devices with MoS₂ since this is a proof-of-concept study, and it is highly time-consuming for us to produce the devices with MoS₂ at this moment.

Quantifying the damage threshold of the device under THz excitation will surely be an important future task. Also, improving the damage threshold will be an important step toward the realization of interesting ultrafast nonlinear events, including e.g. THz-driven phase transitions using the hybrid 3D-2D devices presented in our work.

5. The THz-induced shift is measured indirectly via optical reflectance. Have the authors considered direct electro-optic sampling to probe the local THz field in MoS₂?

Indeed, it would be very interesting to perform the direct EO sampling at the position of MoS₂.

One possible way to perform EO sampling is by making a 2D-3D hybrid THz antenna structure with an EO crystal layer instead of the current spacer and 2D material layer. Let us estimate the longitudinal THz-field amplitude to achieve an observable polarization rotation in a <100>-cut ZnTe. It is known that <100>-cut ZnTe crystal detects the longitudinal THz field with the same efficiency as a <110>-cut ZnTe, a common EO crystal, detects the transverse THz field [A. Nahata, W. Zhu, *Opt. Express*, **15**, 5616–5624 (2007)]. By using ZnTe's material parameters and thickness of 150 nm (same as the combined thickness of both spacer layers in our device), we can estimate that 1 MV/cm longitudinal THz field induces 1 % of the modulation signal. Therefore, we should be able to detect the longitudinal THz field generated in our devices, if it can be loaded with a suitable EO crystal. However, the fabrication method for such a device should be much different from our scheme, and it will be a future task to characterize the THz field with such a device.

One may think about using the MoS₂ flake directly for EO sampling, but it is not realistic due to the extremely small thickness of the MoS₂ monolayer flake. Note that we should use a monolayer flake, which lacks inversion symmetry and thus shows the EO effect. Although we could not find a literature value of the EO coefficient necessary for the detection of a longitudinal field, we can estimate an in-plane THz field needed for the observation of a polarization rotation in monolayer MoS₂. The EO coefficient r_{41} , thickness, and refractive index at 800 nm of MoS₂ are 1.2 pm/V, 0.7 nm and 3.9, respectively [C.-Y. Wang, G.-Y. Guo, *J. Phys. Chem. C Nanomater. Interfaces*, **119**, 13268–13276 (2015)][G. A. Ermolaev et al., *Npj 2D Mater. Appl.* **4**, 1–6 (2020)]. From these parameters, we can calculate the THz field strength on a MoS₂ needed to obtain a measurable 1% modulation signal as 240 MV/cm, which is unrealistically large for our device.

6. Given that THz fields oscillate on a sub-picosecond timescale, does the optical probe's time resolution sufficiently capture the fastest THz-induced changes?

The time resolution of our optical probe would be defined by the 95% confidence band (CB) of 200 - 300 fs for the chirp-compensation curve shown in Supplementary Figure C1. Let us assume that the CB width is the same as the duration of the white-light probe pulse in each frequency bin resolved by the spectrometer. Assuming that the temporal shape of the pulse in each bin is Gaussian, we can obtain the Gaussian width corresponding to the CB width as $\sigma = \Delta_{\text{CB}}/4 = 75$ fs, using the integral of

$$\int_{-2\sigma}^{2\sigma} \frac{1}{\sigma\sqrt{2\pi}} \exp[-(x^2/2\sigma^2)] dx \approx 95.5\%.$$

This corresponds to a full-width half maximum (FWHM) of 176 fs, much smaller than the cycle of THz pulse of approximately 1.5 ps. Therefore, it does not significantly limit our observation.

7. Some technical terms require clearer definitions for interdisciplinary readers. For instance, a simple energy level diagram illustrating the "THz-induced Stark shift" would aid understanding.

We thank the Reviewer for this suggestion. Following the Reviewer's comment, we have added a sub-Figure 4f in the main text to schematically illustrate the THz-induced Stark shift under the built-in bias field, as driven by the THz waveform presented in Fig. 4c.

8. The conclusion could better highlight potential applications, such as ultrafast optical modulators or THz-driven excitonic switches.

We thank the Reviewer for this comment, which is also along the lines of similar comments by the Reviewers 1 and 2. In response to the comments of all 3 Reviewers regarding the field of potential applications of our technology, we have correspondingly expanded the discussion in the Conclusion section, and added several new references. Please also see our response to Reviewers 1 and 2.

9. The supplementary section would benefit from improved clarity. For example, the description "Figure F1. Template named as ***_summary_measTPOPdata, which shows the measured TPOP response of each antenna.

We appreciate the Reviewer for pointing out unclear description in the supplementary file. We modified the corresponding figure captions as follows: "An example of a figure named 'XXX_summary_measTPOPdata,' that shows the measured TPOP response for each antenna. XXX is an identifier for each experiment." In addition, we have modified the first paragraph of Supplementary Section F for more clarity.